

# Trends in atmospheric ammonia at urban, rural and remote sites

# across North America

Xiaohong Yao[1*], Leiming Zhang[2]

[1.] Lab of Marine Environmental Science and Ecology, Ministry of Education,

Ocean University of China, Qingdao 266100, China

[2]Air Quality Research Division, Science and Technology Branch, Environment and

Climate Change Canada, 4905 Dufferin Street, Toronto, Ontario, M3H 5T4, Canada

*Corresponds to: Xiaohong Yao ( xhyao@ouc.edu.cn)





**Abstract.** Interannual variabilities in atmospheric ammonia ($NH_3$) during the most
recent seven to eleven years were investigated at fourteen sites across North America
using the monitored data obtained from NAPS, CAPMoN and AMoN networks. The
long-term average of atmospheric $NH_3$ ranged from 0.8 to 2.6 ppb, depending on
location, at four urban and two rural/agriculture sites in Canada. The annual average
at these sites did not show any deceasing trend with largely decreasing anthropogenic
$NH_3$ emission. An increasing trend was actually identified from 2003 to 2014 at the
downtown Toronto site using either the Mann-Kendall or the Ensemble Empirical
Mode Decomposition method, but "no" or "stable" trends were identified at other
sites. The ~20% increase during the 11-year period at the site was likely caused by
changes in $NH_4^+$/$NH_3$ partitioning and/or air-surface exchange process as a result of
the decreased sulfur emission and increased ambient temperature. The long-term
average from 2008 to 2015 was 1.6-4.9 ppb and 0.3-0.5 ppb at four rural/agriculture
and at four remote U.S. sites, respectively. A stable trend in $NH_3$ mixing ratio was
identified at one rural/agricultural site while increasing trends were identified at three
rural/agricultural (0.6-2.6 ppb, 20-50% increase from 2008-2015) and four remote
sites (0.3-0.5 ppb, 100-200% increase from 2008-2015). Increased ambient
temperature was identified to be a cause for the increasing trends in $NH_3$ mixing ratio
at four out of the seven U.S. sites, but what caused the increasing trends at other U.S.
sites needs further investigation.
**Keywords:** Gas-particle partitioning, reduced nitrogen, Ensemble Empirical Mode





Decomposition, climate impact.
**1. Introduction**
Atmospheric ammonia ($NH_3$) plays an important role in formation of ammonium
sulfate/nitrate aerosols in the size range of nanometer to supermicron (Kulmala et al.,
2004; Ianniello et al., 2011; Yao and Zhang, 2012; Schiferl et al., 2014; Paulot and
Jacob, 2014). The sum of sulfate, nitrate and ammonium ions ($NH_4^+$) usually consist
of the major portion of $PM_{2.5}$ across Canada and the U.S. (Dabek-Zlotorzynskaet al.,
2011; Hand et al., 2012). With significant decreases in acidic gas emissions in the last
decades across North America, e.g., emissions of $SO_2$ and $NO_x$ in Canada decreased
from $2.28 \times 10^6$ and $2.72 \times 10^6$ tones/year in 2003 to $1.23 \times 10^6$ and $2.06 \times 10^6$ tones/year
in                     2013,                    respectively,
(http://www.ec.gc.ca/inrp-npri/donnees-data/ap/index.cfm?lang=En), more attentions
are paid to the relationship between $NH_3$ and $NH_4^+$ aerosols (Zhang et al., 2010; Day
et al., 2012; Nowak et al., 2012; Yao and Zhang, 2012; Schiferl et al., 2014; Zhu et al.,
2013; Markovic et al., 2014; Paulot and Jacob, 2014).

$NH_3$ mixing ratios are affected by several factors such as $NH_3$ emissions, $NH_3/NH_4^+$
partitioning, and meteorological conditions (Sutton et al., 2009; Yao and Zhang, 2013;
Hu et al., 2014). In Europe, previous studies showed that the long-term trend in
atmospheric $NH_3$ observed in some countries didn't show a decrease with a dramatic
decrease in $NH_3$ emissions and the phenomena was referred as "Ammonia Gap"



(Sutton et al., 2009; Ferm and Hellsten, 2012). Long-term trends in atmospheric $NH_3$
across North America are poorly understood (Zbieranowski and Aherne, 2011; Hu et
al., 2104; Van Damme et al., 2014). Such knowledge is important for accurate
prediction of ammonium sulfate/nitrate aerosol levels in the future (Pye et al., 2009;
Walker et al., 2012). In North America, established anthropogenic $NH_3$ emission
inventories showed that agricultural emissions accounted for over 80% of the total
anthropogenic $NH_3$ emissions (Lillyman et al., 2009; Behera et al., 2013; Xing et al.,
2013). However, agricultural emission sources only affect mixing ratios of
atmospheric $NH_3$ at short downwind distances (Theobald et al., 2012; Yao and Zhang,
2013). Non-agriculture emissions such as those from local traffics, waste containers
and soil/vegetation were reported to be important contributors to $NH_3$ in urban
atmospheres (Whitehead et al., 2007; Ellis et al; 2011; Reche et al., 2012; Sutton et al.,
2013; Yao et al., 2013, Hu et al., 2014), although these sources only accounted for a
few percentages of the total $NH_3$ emissions in Canada and the U.S. Due to new
technology adopted, traffic-derived $NH_3$ decreased gradually (Bishop et al., 2010;
http://www.ec.gc.ca/inrp-npri/donnees-data/ap/index.cfm?lang=En). Yao et al. (2013)
and Hu et al (2014) recently reported that the traffic-derived $NH_3$ was a negligible
contributor to atmospheric $NH_3$ in Toronto. However, under climate warming,
soil/vegetation $NH_3$ emissions are expected to increase accordingly, e.g., $NH_3$
volatilization potential nearly doubles under every 5° C increase (Pinder et al., 2012;
Sutton et al., 2013).



Atmospheric NH₃ and ammonium sulfate/nitrate aerosols can be transported
downwind and eventually deposited to natural ecosystems to enhance carbon fixation.
Excessive NH₃ deposition may cause adverse effects such as reduced biodiversity and
eutrophication (Krupa et al., 2003; Erisman et al., 2007; Beem et al., 2010; Bobbinket
al., 2010; Pinderet al., 2012). Recent evidence shows changes in species composition
for sensitive vegetation types at the annual average concentration of 1 μg m$^{-3}$ NH₃
(Cape et al., 2009). Climate warming may increase the vulnerability of ecosystems
towards exposure to NH₃. Thus, trend analysis of atmospheric NH₃ at remote sites will
help to better understand its potential impacts on sensitive natural eco-systems.

In this paper, interannual variabilities in atmospheric NH₃ at fourteen sites across
Canada and the U.S. were investigated, with particular attention paid to its long-term
trends and causes. The fourteen sites include four urban sites, four remote sites and
six rural/agriculture sites distributing at different latitudes. Two trend analysis tools,
i.e., the Mann-Kendall (M-K) analysis (Gilbert, 1987) and the Ensemble Empirical
Mode Decomposition (EEMD, Wu et al., 2009), were used to resolve the time series
of atmospheric NH₃ in mixing ratio at these sites. The analysis results provided new
light on the long-term trends in atmospheric NH₃ at various sites across North
America.

**2. Methodology**
In this study, mixing ratios of atmospheric NH₃ generated at monthly interval were





compiled from three data sources, i.e., the National Air Pollution Surveillance (NAPS,
http://www.ec.gc.ca/rnspa-naps/) network, the Canadian Air and Precipitation
Monitoring Network (CAPMoN), and the Passive Ammonia Monitoring Network
(AMoN, http://nadp.sws.uiuc.edu/nh3Net). Missing data is a common problem during
long-term observations. The sites chosen in this study were based on data
availabilities as detailed below.

The NAPS network is to provide accurate and long-term air quality data across
Canada. At each site, a $PM_{2.5}$ sampler equipped with denuders is used to measure
concentrations of $NH_3$ and acidic gases and particulate chemical components such as
$pNH_4^+$ and $pNO_3^-$ in $PM_{2.5}$. The sampler operates for a 24-hr duration on every third
day. At a few sites, technical problems resulted in $NH_3$ and $pNH_4^+$ concentration data
missing for several months. At four urban sampling sites and one rural/agriculture site
(Fig. 1), the measurements allowed obtaining continuous time series of monthly
averaged concentrations of atmospheric $NH_3$ and $pNH_4^+$ and were thereby used for
trend analysis. However, these sites also suffer from the problem of missing data. For
example, there was only ~70% months when 8-10 sets of 24-hr data were available to
calculate the monthly average value. In a few months, there were only 1-3 sets of
24-hr data available to do so. This may cause uncertainty on the calculated trends in
atmospheric $NH_3$ and $pNH_4^+$. Moreover, one site at Egbert in the southern Ontario
(Fig. 1), the part of CAPMoN, also had the long-term measurement concentrations of
atmospheric $NH_3$ and $pNH_4^+$ using the identical sampler as used in the NAPS network.





The site is located at a rural/agriculture area. The data was also averaged monthly for
the trend analysis. The six Canadian sites were referred as Sites 1-6 on basis of their
annual average mixing ratios of atmospheric $NH_3$ in decreasing order.

The AMoN within the National Atmospheric Deposition Program in the U.S. started
operation in fall 2007. An important objective of AMoN is to assess long-term trends
in $NH_3$ concentrations and its deposition. AMoN included only sixteen sites in 2007
and dozens of sites are now available. The Radiello® passive samplers are deployed
every two weeks at each site according to the standard operating procedure for
monitoring atmospheric $NH_3$. Puchalski et al. (2015) recently compared the bi-weekly
passive measurements with those measured by annular denuder systems (ADS) at
several AMoN sites and found that the mean relative percentage difference between
the ADS and AMoN sampler was -9%. In this study, mixing ratios of atmospheric
$NH_3$ at eight AMoN's sites were selected for the trend analysis on basis of two criteria
(Fig 1): 1) the length of the valid data should be at least seven years according to
Walker et al., (2000); and 2) there were no monthly average data missing in each year.
The eight sites were refereed as Site 7-14 (Fig 1), four of which were located at rural
areas and another four at remote areas. Consistent with the six sites across Canada, the
monthly averages at the eight sites were used for data analysis if not specified.
Moreover, all ambient temperature (T) data were obtained from on-site records or
nearby meteorological stations.



The M-K analysis is a non-parametric statistical procedure which can be used to
analyze trends in data sets including irregular sampling intervals, data below the
detection limit, and trace or missing data (Kampata et al., 2008). Considering the data
flaws aforementioned were indeed presented in our selected datasets to different
extents, the M-K analysis is thereby used to resolve the time series of the annual
average of $NH_3$ in this study. The M-K method yields qualitative trend results such as
"increasing/decreasing", "probable increasing/decreasing", "stable" and "no trend",
depending on the calculated "S" statistic, confidence factor and coefficient of
variation (Gilbert, 1987). Moreover, the EEMD is a recently developed statistical tool
to determine the trend of a time series of a variable in various fields such as
economics, health, environment and climate (Wu et al., 2009). The EEMD built on
Empirical Mode Decomposition (EMD) and was updated by Wu et al. (2009) to
overcome the problem of mode mixing in the EMD. The method has since been
applied widely (e.g., Erturk et al., 2103; Ren et al., 2014) because it is most suitable
for resolving non-stationary and non-linear signals. The mixing ratio of atmospheric
$NH_3$ was affected not only by its emissions, atmospheric transport, dilution and
deposition, but also affected by non-stationary and non-linear chemical reactions
(Ianniello et al., 2011; Hu et al., 2014). Thus, the EEMD is also used in this study and
is briefly introduced here.
In general, all data are amalgamations of signal and noise as shown below:
$X(t)=S(t)+N(t)$
where $X(t)$ is the record data, and $S(t)$ and $N(t)$ are the true signal and noise,



respectively. In the EMD, any dataset is assumed to consist of different simple
intrinsic modes of oscillations. Each of these intrinsic oscillatory modes is represented
by an intrinsic mode function (IMF). In the EEMD, white noise is added to the single
data set, X(t), and the ensemble mean is used to improve accuracy of measurements.

**3. Results and discussion**


*3.1 Temporal variations of atmospheric NH₃ at the six Canadian sites*
Fig. 2 shows monthly variations of atmospheric $NH_3$ in mixing ratio at the six
Canadian sites. At Site 1, an urban site in downtown Toronto, the measured mixing
ratios were 2.6±1.2 ppb (average ± standard deviation) during the period from July
2003 to June 2014 (Fig 2a and Table 1). When compared with those reported in other
urban atmospheres, the long-term average value of 2.6 ppb ranked at a moderately
low concentration level (Whitehand et al., 2007; Saylor et al., 2010; Alebic-Juretic,
2008; Meng et al., 2011). Interannual variations were evident at Site 1 with the
coefficient of variation (CV) of 0.11, defined as the ratio of the standard deviation to
the average. It should be noted that the annual averages in 2004 and 2005 were
calculated from July 2003 to June 2004 and from July 2004 to June 2005, respectively,
instead of a calendar year, in order to obtain the longest time series of annual averages.
The similar calculations were used for other years and other sites. The M-K analysis
result suggested an increasing trend from 2004 to 2014 with a confidence level of
98%. When intra-annual variations were analyzed at Site 1, a distinctive seasonal
trend was obtained with the highest seasonal average value of 3.7±0.7 ppb in summer





(June to August) and the lowest of 1.3±0.6 in winter (December to the next February).

The M-K analysis results showed either "no" or "stable" trends in atmospheric NH$_3$ at
the other Canadian sites with long-term average of 2.4±0.6 ppb at Site 2, 2.1±1.2 ppb
at Site 3, 1.9±0.8 ppb at Site 4, 1.6±0.5 ppb at Site 5, and 0.8±0.6 ppb at Site 6 (Fig.
2b-f). However, interannual variations at these sites were evident, e.g., the CV values
calculated from annul averages varied from 0.07 to 0.19, depending on location
(Table 1). Atmospheric NH$_3$ at Sites 3, 4 and 6 exhibited a distinctive seasonal
variation, but this was not the case at Sites 2 and 5. The largest seasonal variation
occurred at Site 3 while the smallest occurred at Site 2. The two sites were selected as
examples for further discussion below.

Site 3 is situated at a rural/agriculture area in Saint-Anicet of Quebec. The largest
seasonal average value occurred in the fall during the measurement period of
September 2003 - August 2014. Fertilizer application usually leads to a sharp increase
in atmospheric NH$_3$ mixing ratio (Lillyman et al., 2009; Yao and Zhang, 2013) and
this was indeed observed at Site 3. For example, there was usually one 24-hr sample
in October having a mixing ratio 1-2 orders of magnitude higher than other samples
collected before or after (figure not shown). The on-site sampling was performed
every third day, and strong NH$_3$ emissions associated with fertilization application
generally occurred within the initial 3-5 days   (Lillyman et al., 2009). Thus,
extremely high mixing ratios could be observed on one day in October in some years,





but not in every year. The episodes further led to large interannual variations with the
value of CV reached 0.19.

Site 2 is located at an urban area in Edmonton. Mixing ratios of atmospheric $NH_3$
were 2.4±0.6 ppb and the differences between seasonal average values were only
0.1-0.3 ppb during the period of May 2006 - April 2014. However, the seasonal
average temperature of 16.7±1.9°C in summer was much higher than that of
-19.8±4.5 °C in winter. Soil/vegetation $NH_3$ emissions should be negligible in such
cold winters, however, industrial and/or non-industrial anthropogenic sources might
be enhanced in winter, which could explain the small seasonal variations in $NH_3$
mixing ratio at this site. This hypothesis was supported by the much higher (2.0-4.0
times) mixing ratios of $SO_2$, HONO and $HNO_3$ in winter as compared to those in
summer (figure not shown).

*3.2 Temporal variations of atmospheric NH₃ at the eight American sites*
For the eight AMoN's sites in the U.S., the data measured from August 2008 to July
2015 were used for analysis at all the sites except at Site 12 for which the data
measured during the period of September 2008 - August 2015 was used (Fig. 3 and
Table 1). Site 7 is located at an intensive agriculture activity zone in Randall of Taxes,
and Sites 8-10 are located at rural areas in Dodge of Wisconsin, Wayne of Michigan,
and Champaign of Illinois, respectively, with moderately intensive agriculture
activities. Long-term average of $NH_3$ was as high as 4.9±1.2 ppb at Site 7 where the





seasonal average in summer was approximately 20% higher than those in the other
seasons. The M-K analysis result showed an increasing trend at this site with a
confidence level of 99.9%. Long-term average of $NH_3$ at Sites 8-10 were 2.6±1.4,
2.2±1.0 and 1.6±1.0 ppb, respectively, and distinctive seasonal variations were seen at
the three sites with the lowest values in winter. The M-K analysis results showed an
increasing trend at Sites 9-10 with a confidence level of >98% and no trend at Site 8.

Sites 11-14 are located at the remote areas in Tompkins of New York, Lake of
Minnesota, Charleston of South Carolina, and Rio Arriba of New Mexico,
respectively. The long-term average $NH_3$ was only 0.3-0.5 ppb at these four remote
sites, but with distinctive seasonal variations with the highest in summer and the
lowest in winter (Table 1). The M-K analysis results showed an increasing trend at the
four sites with a confidence level of >95%.

*3.3 Exponential correlations between $NH_3$ and T*
When local soil/vegetation emissions were the major contributors to atmospheric $NH_3$,
its mixing ratio usually exhibited as an exponential function of ambient T (Sutton et
al., 2009; Flechard et al., 2013; Hu et al., 2014). Thus, the exponential correlation
relationship was examined at the fourteen sites to identify potential major contributors
to atmospheric $NH_3$. Note that a perfect exponential correlation with $R^2 > 0.9$ would
occur only when the soil/air mass transfer of $NH_3$ was not the limitation factor
(Flechard et al., 2013; Hu et al., 2014), and soil/air mass transfer rate associated with



dry soil was much small (Su et al., 2011).

A moderately good exponential correlation was obtained at Site 1 with $R^2$=0.74 and P
value <0.01 (Fig. S1a). $NH_3$ emissions from green space surrounding this site likely
played a major role in the observed $NH_3$ level (Hu et al., 2014). Similar results were
obtained at Sites 3, 4 and 6 when a few exterior data points were excluded. For
example, five data points at Site 3 severely deviate from the regression curve because
of fertilizer application (Fig. S1c). When these five data points were excluded, $R^2$
reached 0.60 and P value <0.01. In addition, $R^2$ was 0.75 at the rural Site 6 when one
outlier data point was excluded (Fig. S1f). The two parameters in the regression
equation $[NH_3]=0.24*exp(0.094*T)$ were largely different from those obtained at the
two    downtown    sites,    i.e.,    $[NH_3]=1.48*exp(0.048*T)$    at    Site    1    and
$[NH_3]=1.29*exp(0.036*T)$ at Site 4, noting that the parameters were close between
the two downtown sites. Under the condition of T below or close to 0°C, the mixing
ratios of atmospheric $NH_3$ at the rural Site 6 were almost one order of magnitude
smaller than those at the downtown sites, leading to the large difference for
parameters in regression equations between the rural and urban sites. The higher
mixing ratio under freezing condition at the Toronto downtown site was proposed to
be likely associated with $NH_3$ emissions from green space (Hu et al., 2014). Flechard
et al (2013) also reported a higher $NH_3$ emission from grassland under freezing
condition, but the corresponding mechanism is still not clear.



At Site 2, the exponential correlation was poor even with two outlier samples being
excluded (Fig. S1b), implying that local soil/vegetation emissions were less likely the
major contributors to atmospheric $NH_3$. Like Site 2, $R^2$ was only 0.39 at Site 5 even
with two exterior data points being excluded (Fig. S1e). $NH_3$ in urban atmospheres
were reported to come from various sources (Whitehead et al., 2007; Ianniello et
al.,2010; Saylor et al., 2010; Meng et al., 2011; Reche et al., 2012), some of which
were less dependent on ambient T.

$R^2$ between atmospheric $NH_3$ and ambient T was below 0.1 at Site 7 (Fig S2a).
Considered the high mixing ratios observed at the rural/agriculture site, it can be
confirmed that local agriculture emissions were the major contributor to atmospheric
$NH_3$ and the agriculture emissions appeared to be independent on ambient T. $R^2$ of
0.64, 0.69 and 0.45 at Sites 8-10, respectively, (Fig. S2b-d) suggested that local
soil/vegetation emissions should be among the major contributors to atmospheric $NH_3$.
The same can be said for the remote Site 11 with $R^2$ of 0.63 (Fig. S2e). $R^2$ was 0.25,
0.2 and 0.15 at remote Sites 12-14 (Fig S2f-i), respectively. When the data measured
in calendar year 2011, 2012, 2013 and 2014 at Site 13 were used for correlation
analysis, respectively; the values of $R^2$ were 0.47 in 2011, 0.58 in 2012, 0.69 in 2013
and 0.74 in 2014. Local soil/vegetation emissions might be among the major
contributors to atmospheric $NH_3$ at the site while the low $R^2$ values in 2011-2012
could be due to analytical errors. In fact, the mixing ratios at the site in 2011-2012
were generally close to the detection limit. A similar calculation was conducted at Site





14 with $R^2$ still below 0.2 in different calendar years, suggesting that local
soil/vegetation emissions were unlikely the major contributors to atmospheric $NH_3$.
Yao and Zhang (2013) proposed that long-range transport could be an important
contributor to atmospheric $NH_3$ at remote sites in North America. When a similar
calculation was conducted at Site 12, $R^2$ was 0.57 in 2012, 0.81 in 2013 and 0.39 in
2014. Local soil/vegetation emissions were possibly among the major contributors to
atmospheric $NH_3$ at the site in 2012 and 2013, but the long-range transport together
with local soil/vegetation emissions might be important contributors to atmospheric
$NH_3$ in 2014.

*3.4 Cause analysis of trends in atmospheric $NH_3$ at Canadian sites*
Site 1 is located in Downtown Toronto, Ontario. Fig S3a shows the annual
anthropogenic $NH_3$ emissions from 2003 to 2013 in Ontario, Canada. Not only the
total $NH_3$ emissions, but also emissions from the four major sectors including
agriculture, mobile, industrial and non-industrial generally decreased. However, an
increasing trend in annual average $NH_3$ was found at Site 1, which was identified to
be caused by (1) the increased T, and (2) the decreased $SO_2$ emission. Increasing T not
only increases soil/vegetation $NH_3$ emissions but also affects $NH_3/pNH_4^+$ partitioning,
both processes would increase $NH_3$ mixing ratios. The decreased $SO_2$ emissions due
to the tightened emission control policies since 2008 by the city and provincial
governments led to significant declines in $SO_2$ oxidation products (Hu et al., 2014;
Pugliese et al 2014), which in turn also affected $NH_3/pNH_4^+$ partitioning and



increased $NH_3$ mixing ratios. These hypotheses were supported by the trends in T and
$pNH_4^+$ and their correlations with that in $NH_3$, as detailed below.

A moderately good correlation ($R^2$=0.76, P value <0.01) was obtained between the
annual average $NH_3$ and the annual average T, while a negative correlation ($R^2$=0.40
and P value <0.05) was obtained between the annual average $NH_3$ and $pNH_4^+$ (Fig.
4a). Note that a decreasing trend in annual average $pNH_4^+$ was found with a
confidence level of >99% based on M-K analysis. When ambient T was increased by
$5^o$ C, the mixing ratio of atmospheric $NH_3$ was increased by ~1 ppb according to the
regression equation.

Fig. S4 shows the intrinsic mode functions (IMFs) and residuals solved by the EEMD
at Site 1. The extracted residuals represented the long-term trend in atmospheric $NH_3$
and the IMFs represented other fluctuations in different time scales. The
EEMD-extracted long-term trend in atmospheric $NH_3$ was generally increased by ~20%
from 2003 to 2014. The EEMD was also used to extract the long-term trend in $pNH_4^+$
in $PM_{2.5}$ from 2003 to 2014 (Fig. S5). Correlation between the two EEMD-extracted
long-term trends resulted in a regression equation of $[NH_3]$ = -1.41*$[pNH_4^+]$ + 4.3,
with $R^2$=0.93 and P value <0.01 (Figure 4b). The unit of $NH_3$ is in ppb while the unit
of $pNH_4^+$ is in $\mu g\ m^{-3}$. The absolute value of the regression slope was almost the same
as the unit conversion coefficient. Thus, the EEMD-extracted long-term trend in
atmospheric $NH_3$ seemed to be mainly determined by the change in $NH_3/pNH_4^+$





partitioning. The increasing T further enhanced this trend. When the EEMD-extracted
long-term trend in ambient T was correlated to that of atmospheric $NH_3$, we obtained
$[NH_3] = 0.13*T+1.5$, $R^2=0.47$ and P value $<0.01$ (Fig 4b). The EEMD-extracted
results suggest that the changes in $NH_3/NH_4^+$ partitioning is one of the dominant
factors influencing the long-term $NH_3$ trend at Site 1. The relative importance
between (1) changes in $NH_3/NH_4^+$ partitioning and (2) increased biogenic $NH_3$
emission due to increasing T is yet to be investigated.

At Site 2, the EEMD-extracted residual of atmospheric $NH_3$ varied within a very
small range (~10%, Fig. 5), which was consistent with stable trend generated from the
M-K analysis. Little correlation was found between the annual average $NH_3$ and T
($R^2<0.05$) or $pNH_4^+$ ($R^2<0.01$). Thus, the $NH_3$ trend identified at Site 2 was seemingly
unaffected by changes in $NH_3/NH_4^+$ partitioning and T-dependent biogenic $NH_3$
emission, or additional local factors cancelled out the impact from the two factors.

Site 3 is a rural/agriculture site and annual agriculture $NH_3$ emissions in Quebec were
stable from 2003 to 2009 with CV of 0.02 (Fig S3b). During the same period, mobile,
industrial and non-industrial emissions were decreased by 40% in Quebec. While the
M-K analysis result showed no consistent long-term trend in atmospheric $NH_3$, the
EEMD-extracted a bell-shaped pattern (Fig. 5), i.e., $NH_3$ increased from 1.7 ppb in
2003 to 2.5 ppb in 2009 and then decreased down to 1.3 ppb in 2014. The
anthropogenic $NH_3$ emission data from 2003 to 2009 didn't support the increasing





trend in atmospheric $NH_3$ at this site during this period.

A good correlation was found between the EEMD-extracted long-term trends in $NH_3$
and T with a linear regression relationship of $[NH_3] = 0.39*T - 0.30$, with $R^2=0.80$
and P value <0.01. The slope of 0.39 was consistent with that reported by Sutton et al
(2013), i.e., $NH_3$ volatilization potential nearly doubles every 5ºC. On the contrary,
little correlation was found between the EEMD-extracted residuals for atmospheric
$NH_3$ and $pNH_4^+$ ($R^2<0.1$), suggesting that the $NH_3/pNH_4^+$ partitioning likely played a
negligible role on the long-term trend in atmospheric $NH_3$ at this site. The increasing
trend in $NH_3$ should mainly be caused by increased biogenic $NH_3$ emission due to the
increased T.

A similar conclusion could also be generated for Site 4 to that for Site 3. The M-K
analysis results showed a stable trend in atmospheric $NH_3$ and a slightly decreasing
trend in $pNH_4^+$ with a confidence level of >99%. The EEMD-extracted long-term
trend showed that $NH_3$ decreased by ~5% from 2007 to 2008 and then increased by
~50% afterwards. Although the correlation between the annual averages $NH_3$ and T
was not very good ($R^2=0.39$, P=0.13), correlation between the EEMD-extracted
long-term trends in $NH_3$ and T was almost perfect ($R^2= 0.90$, P<0.01). No correlation
was found between the annual average $NH_3$ and $[pNH_4^+]$ ($R^2 <0.01$), and a relatively
low correlation was found between the EEMD-extracted long-term trends in
atmospheric $NH_3$ and $pNH_4^+$ ($R^2=0.39$, P value <0.01). These results suggested that



the long-term change in ambient T possibly dominated the long-term trend in
atmospheric $NH_3$ at the site.

Site 5 is an urban site located in British Columba, Canada. Anthropogenic $NH_3$
emissions were decreased from 2003 to 2013 in this province (Fig S3c). The M-K
analysis result showed no trend in atmospheric $NH_3$ at Site 5 from 2003 to 2014 and
the EEMD-extracted long-term trend was almost constant. Based on the correlation
analysis of the EEMD-extracted results (not shown), the long-term changes in
ambient T and $pNH_4^+$ cannot explain the EEMD-extracted trend in atmospheric $NH_3$
at this site.

At Site 6, the EEMD-extracted trend in atmospheric $NH_3$ showed an increase of ~10%
from 2003 to 2006 and then a decrease of ~40% afterwards (Fig. 5). Poor correlations
were found between annual average $NH_3$ and T or between $NH_3$ and $NH_4^+$ with P
values >0.05. Meaningless correlations between the EEMD-extracted trends in $NH_3$
and T or between the EEMD-extracted trends in $NH_3$ and $NH_4^+$ were obtained. The
trend in T and $NH_3/NH_4^+$ partitioning cannot explain the long-term variations of
atmospheric $NH_3$.

*3.5 Cause analysis of trends in atmospheric $NH_3$ at the U.S. sites*
At the eight U.S. sites, $R^2$ between annual average $NH_3$ and T were all below 0.2 with
P values all larger than 0.1. The simple correlation analysis did not provide direct



evidence that T was the dominant factor affecting the NH₃ trend. However, the
EEMD-extracted trends in NH₃ and T had a much better correlation at some sites, e.g.,
with $R^2$= 0.85, 0.99, 0.54 and 0.99 at Site 7, 8, 9 and 13, respectively, and with P
values small than 0.01. Thus, the increasing T should be one of the main factors
causing the long-term trend in NH₃ at these four sites. Note that no reasonable
relationship was identified between trends in NH₃ and T at the other four sites using
the EEMD method.

The EEMD-extracted long-term trend showed an increase in atmospheric NH₃ from
4.2 ppb in August 2008 to 6.8 ppb in July 2015 at Site 7 (Fig. 6a), from 2.4 ppb in
August of 2008 to 3.0 ppb in July of 2015 at Site 8 (Fig. 6b), from 1.8 ppb in August
of 2008 to 2.8 ppb in July of 2015 at Site 9 (Fig. 6c), a complex varying pattern
during the period from August 2008 to July 2015 at Site 10 (Fig. 6d), and an
increasing trend (by 0.3-0.5 ppb, or 100-200% in percentages) at Sites 11-14 (Fig.
6e-h). The percentage increases (100-200%) in NH₃ mixing ratio from 2008 to 2015
at the remote sites were substantially larger than those at the rural/agriculture sites

413   (20-50%).


NH₃ emissions in the United States increased by 11 % during the period from 1990 to
2010 due to the growth of livestock activities (Xing et al., 2013). This is particularly
the case in North Carolina and Iowa. This increase along is not enough to explain the
~50% increase in NH₃ from 2008 to 2015 at Site 7 which is located in Taxes. The





increased T is believed to be another important factor causing the increased NH₃ at
this site as mentioned above. It is also noted that the increasing trends in NH₃ at Site
11-14 identified using the EEMD-extracted results were also consistent with the M-K
analysis results.

The annual average NH₃ at the remote sites reached 0.4-0.6 ppb in 2015. Assuming
the same increasing rate continues for another 7-10 years, the annual average will
exceed the proposed critical level of 1 μg m⁻³ at two sites for protecting sensitive
ecosystems (Cape et al., 2009).

**4. Conclusions**
Long-term average of atmospheric NH₃ was in the range of 0.3-0.5, 1.6-2.6, and
0.8-4.9 ppb at the remote, urban, and rural/agriculture sites, respectively, across the
North America. Moderate exponential correlations between atmospheric NH₃ and
ambient T were found at nine sites, implying that local biogenic emissions and/or
NH₃/NH₄⁺ partitioning were likely dominant factors causing the long-term trends in
atmospheric NH₃ at these sites.

At the four Canadian sites, no decreasing trends in atmospheric NH₃ were found
despite significant decreases in anthropogenic NH₃ emissions from main sectors in the
last decade. The decreased NH₃ anthropogenic emission was compensated or
overwhelmed by the increased biogenic emission and/or changes in NH₃/NH₄⁺





partitioning. This was supported by $pNH_4^+$ data which exhibited a decreasing trend,
likely caused by a combination of reduced $SO_2$ and $NO_x$ emission and increased
temperature. No decreasing trends in atmospheric $NH_3$ were found at other two
Canadian sites, but it was unknown what caused this.

The M-K analysis showed an increasing trend in atmospheric $NH_3$ at seven out of the
eight U.S. sites, which was also supported by the EEMD-extracted results. $NH_3$
increased by 20-50% from 2008 to 2015 at the three rural/agriculture sites and by
100%-200% at the four remote sites. If the same increasing trend continues in the next
5-7 years, the annual average $NH_3$ at two remote sites will exceed 1 μg m$^{-3}$, a level
below which has been proposed to protect sensitive eco-systems at the remote sites.

In most cases, the two statistical approaches used in the present study yield consistent
trends in atmospheric $NH_3$ measured at different sites. The EEMD method appeared
to have more powerful interpretation ability for resolving trends because 1) it is less
affected by extremely high concentration points, and 2) it yields a continuous and
quantitative trend result. However, this method occasionally suffers from "the end
effect" and leads to physically meaningless results. Using the combined (or more than
one statistical methods) can better resolve and interpret long-term trends in
atmospheric $NH_3$.

**Acknowledgement**



Ammonia Monitoring Network (http://nadp.sws.uiuc.edu/data/sites/list/?net=AMoN)
is acknowledged for downloading data for analysis. The work is financially supported
by the Clear Air Regulatory Agenda of Canada and xhy thanks the support from the
National Program on Key Basic Research Project (973 Program: 2014CB953700) of
China.

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





Table 1. The mixing ratios of atmospheric $NH_3$ at fourteen sites ($NH_3$ unit in ppb, T unit in $^0C$, Sites 1-14 were defined in the text)

| Site | Sampling Period | Annual $NH_3$ | Spring $NH_3$ | T | Summer $NH_3$ | T | Fall $NH_3$ | T | Winter $NH_3$ | T |
|------|-----------------|---------|--------|---|--------|---|------|---|--------|---|
| 1 | July. 2003-June. 2014 | 2.6±1.2 | 2.8±1.2 | 8.0±6.0 | 3.7±0.7 | 21.6±1.5 | 2.8±0.7 | 11.8±5.7 | 1.3±0.6 | -2.5±3.1 |
| 2 | May. 2006-Apr. 2014 | 2.4±0.6 | 2.3±0.6 | 3.7±7.4 | 2.4±0.4 | 16.7±1.9 | 2.6±0.6 | 4.2±7.7 | 2.3±0.9 | -19.8±4.5 |
| 3 | Sep. 2003-Aug. 2014 | 2.1±2.0 | 1.6±0.8 | 6.1±6.7 | 3.0±1.5 | 19.5±2.2 | 3.2±2.8 | 9.1±5.6 | 0.5±0.3 | -6.8±2.9 |
| 4 | Nov. 2007-Oct. 2014 | 1.9±0.8 | 1.7±0.7 | 8.3±6.9 | 2.7±0.5 | 21.5±2.1 | 2.1±0.4 | 9.4±6.3 | 1.0±0.2 | -6.1±2.6 |
| 5 | Sep. 2003-Aug. 2014 | 1.6±0.5 | 1.5±0.4 | 10.1±2.9 | 1.9±0.4 | 17.8±1.9 | 1.7±0.5 | 10.6±4.1 | 1.4±0.4 | 4.2±1.5 |
| 6 | Aug. 2003-July 2011 | 0.8±0.6 | 1.0±0.7 | 6.7±5.8 | 1.2±0.4 | 19.4±2.2 | 0.7±0.3 | 9.7±5.3 | 0.2±0.2 | -5.6±2.8 |
| 7 | Aug. 2008-July. 2015 | 4.9±1.2 | 4.6±1.2 | 14.2±4.2 | 5.5±0.8 | 25.5±1.7 | 4.7±1.5 | 14.7±5.3 | 4.6±1.4 | 3.0±1.8 |
| 8 | Aug. 2008-July. 2015 | 2.6±1.4 | 3.2±1.2 | 7.4±6.5 | 3.8±0.9 | 20.7±1.8 | 2.5±0.6 | 9.3±5.9 | 1.0±0.5 | -7.4±3.8 |
| 9 | Aug. 2008-July. 2015 | 2.2±1.0 | 2.6±1.0 | 10±6.1 | 3.2±0.7 | 22.4±1.7 | 2.1±0.7 | 11.5±5.5 | 1.1±0.3 | -3.2±3.3 |
| 10 | Aug. 2008-July. 2015 | 1.6±1.0 | 2.3±1.0 | 11.3±5.7 | 1.9±0.4 | 22.1±1.5 | 1.8±0.8 | 11.2±5.6 | 0.4±0.4 | -3.3±3.3 |
| 11 | Aug. 2008-July. 2015 | 0.5±0.4 | 0.7±0.4 | 6.6±6.5 | 0.8±0.3 | 18.4±1.5 | 0.3±0.2 | 9.2±5.0 | 0.1±0.1 | -4.9±3.4 |
| 12 | Sep. 2008-Aug. 2015 | 0.3±0.3 | 0.3±0.2 | 3.4±6.1 | 0.5±0.3 | 17.5±2.6 | 0.3±0.3 | 5.5±6.7 | 0.1±0.1 | -13±4.5 |
| 13 | Aug. 2008-July. 2015 | 0.3±0.4 | 0.2±0.2 | 11±3.6 | 0.7±0.5 | 23.8±1.5 | 0.2±0.2 | 12.5±6.0 | 0.2±0.1 | 0.2±2.3 |
| 14 | Aug. 2008-July. 2015 | 0.3±0.3 | 0.4±0.2 | 18.6±4.2 | 0.5±0.3 | 27.8±1.0 | 0.2±0.2 | 19.7±4.8 | 0.2±0.2 | 10.3±3.2 |







**List of Figures**


Fig. 1 Map of fourteen long-term atmospheric $NH_3$ monitoring sites across North America.

Fig. 2 Monthly averages of atmospheric $NH_3$ measured at six Canadian sites.

Fig. 3 Monthly averages of atmospheric $NH_3$ measured at eight U.S. sites.

Fig. 4 Correlations between atmospheric $NH_3$ and ambient T at Site 1 (a: annual average value; b: EEMD-extracted trend).

Fig. 5 The long-term trends in atmospheric $NH_3$ extracted by the EEMD at six Canadian sites.

Fig. 6 The long-term trends in atmospheric $NH_3$ extracted by the EEMD at eight U.S. sites.








Fig 1

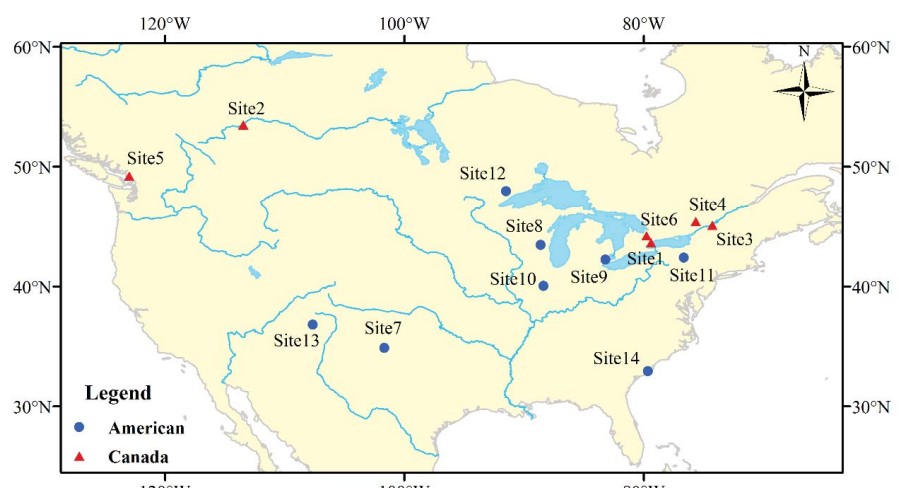

















Fig 2

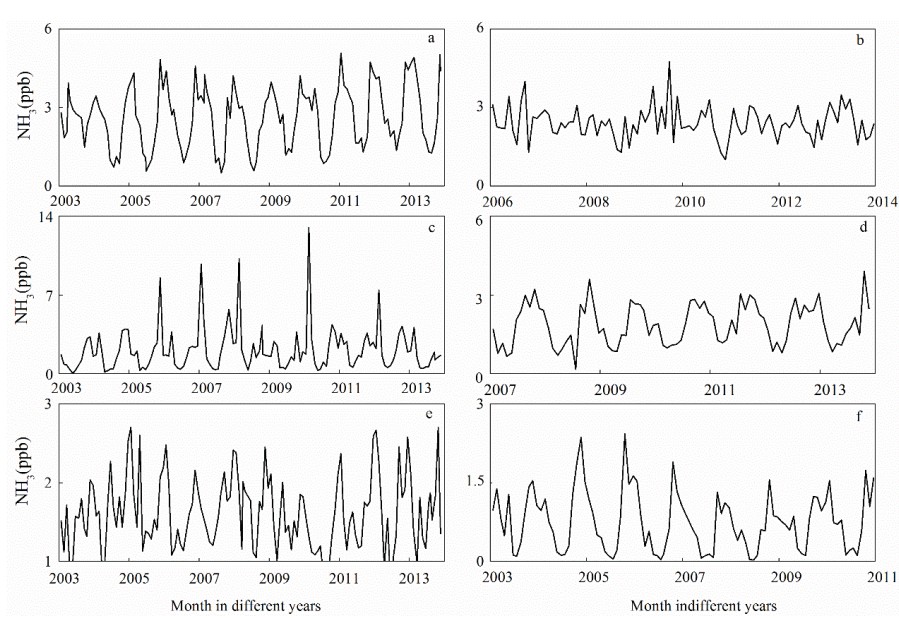














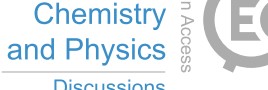


Fig 3

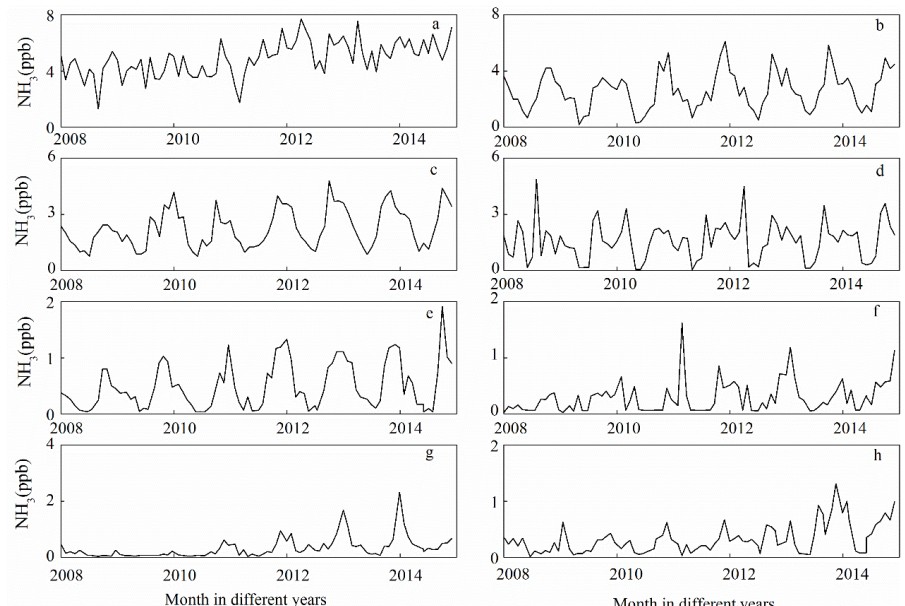
















Fig 4

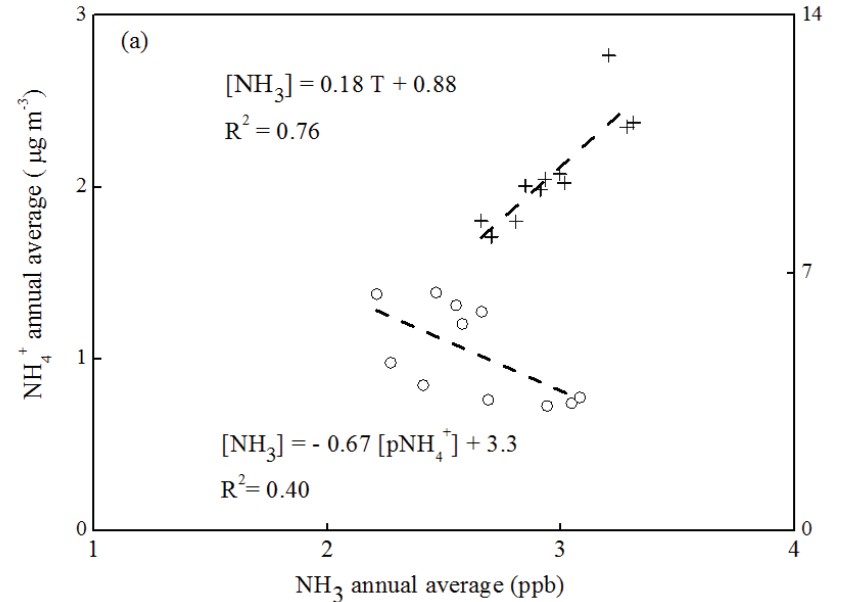


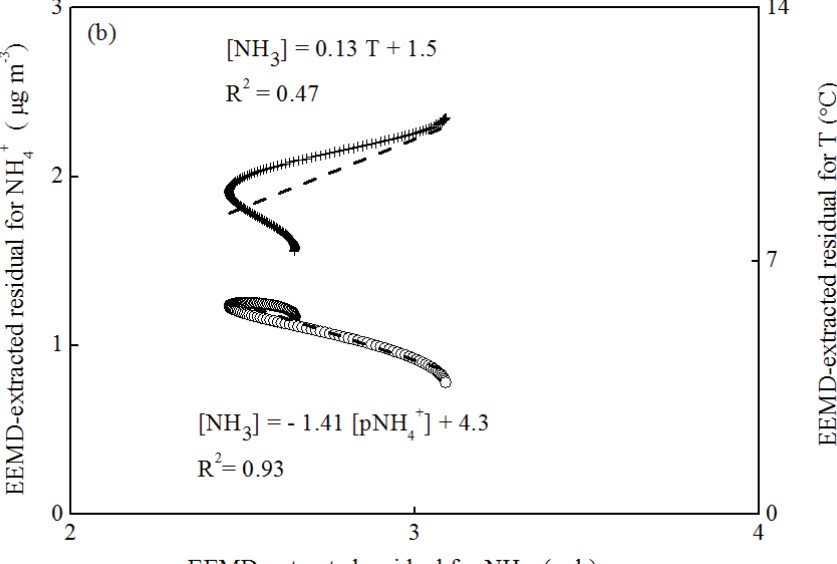









Fig 5

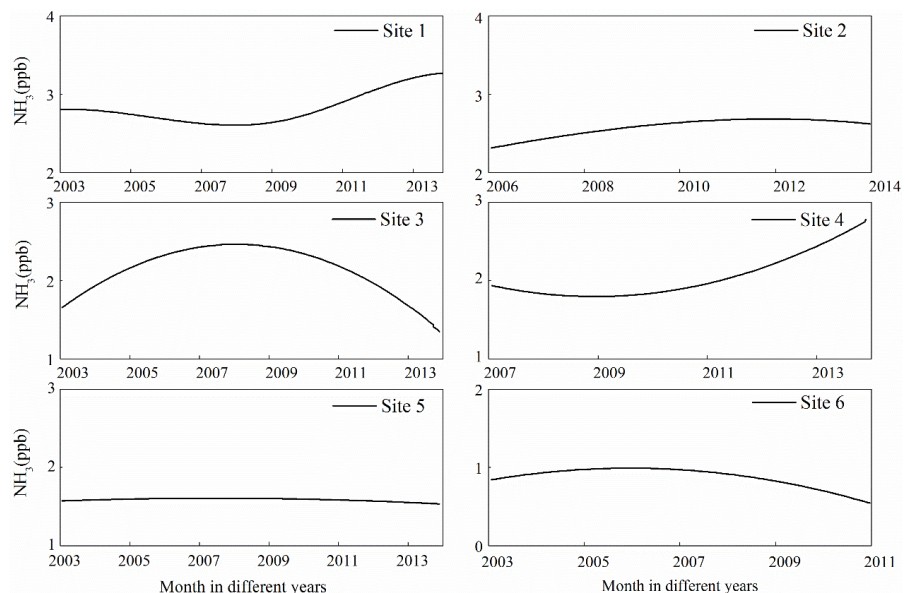















Fig 6

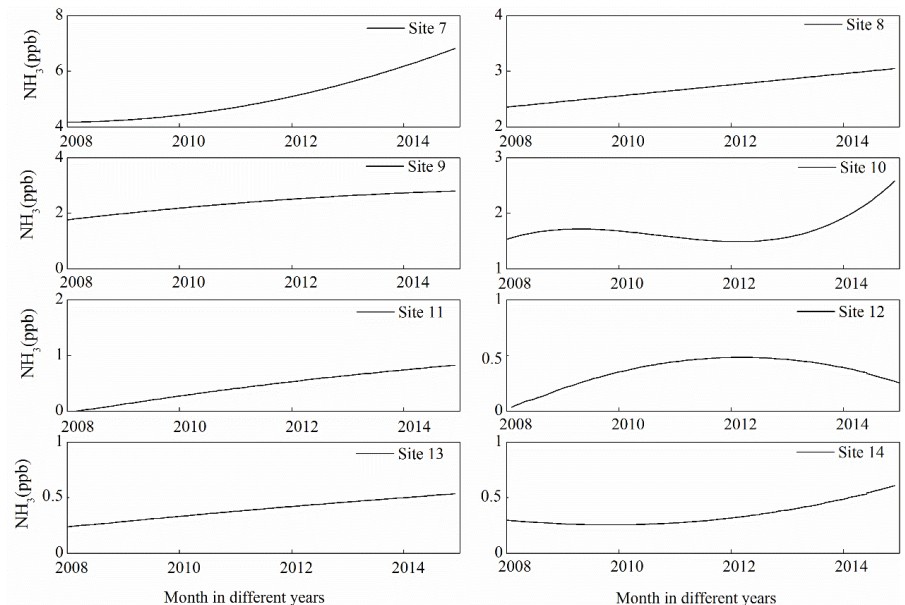



