# Peer review of "Fig S1. Exponential correlations between atmospheric $\text{NH}_3$ and ambient T at six Canadian sites"

_Atmospheric Chemistry and Physics, 2016_

## Referee Comment (RC1) · C. Flechard (Referee) · 3 Jun 2016

**Reviewer's comments on ACPD-2016-259 manuscript "Trends in atmospheric ammonia at urban, rural and remote sites across North America" by X. Yao and L. Zhang**

**General comments**

The paper presents an analysis of temporal trends, and to some extent spatial patterns, of long-term (>10 years) ambient atmospheric ammonia (NH3) concentrations measured across Canadian and US air pollution monitoring networks. Temporal trends are compared between a selection of 14 urban, rural and remote sites by using statistical trend analysis tools (Mann-Kendall, M-K and Ensemble Empirical Mode Decomposition, EEMD). The paper provides a useful and original study of NH3 trends at the N. American continental scale, and fits well within the scope of Atmospheric Chemistry and Physics, even though one may deplore the fact that not all sites and data available in the networks were analyzed.

Long-term trends are interpreted in terms of changing emission patterns and changing pollution climate and temperature, but some of the arguments and hypotheses are less than compelling. In particular the argument that increasing NH3 concentrations at some sites may be explained by a significant upward trend in mean temperatures on such short time scales - from a climatological viewpoint (only 10 years) - does not sound convincing, especially since no long-term temperature data are shown alongside the NH3 concentration time series.

Generally the paper could be improved by a better description of the methods used, both in terms of measurement techniques and statistical methods, and the figures should be re-arranged to combine the actual measured time series with the trends analysis to better illustrate the arguments.

**Specific comments**

Methods

p6, l96-100: please provide more details of the sampling and measurement techniques used in NAPS and CAPMON: which PM2.5 sampler is used (name/manufacturer), are the denuders wet or dry, what is the sample flow rate, are the data hourly or daily integrated values, how is NH4+ measured in the lab after extraction, or is it in-situ online analysis, etc...??

p6, l100-107: I agree that missing data are an important problem when dealing with the analysis of long term temporal trends, especially if the downtime periods are not randomly distributed but might tend to coincide with specific weather patterns, eg very cold or very wet, etc. Thus it would be useful to indicate the monthly/annual data capture rates (eg number of days per month of available data, or rates of missing data, whichever ) alongside the measured concentrations in Figs. 2-3, on a separate axis with a different color or symbol. For example show the missing data rate as a vertical bar for each month, so the figure wouldn't be too cluttered.

p8, l133-141: please provide very briefly the mathematical basis of the statistical method (in which way

does it differ from a parametric procedure?)

Results and discussion

p10, l189-193: is there any actual evidence from on-site observations that fertilization takes place in the fall at or around Site 3? Why should there be any fertilizer application after harvest and just before winter, when there is no longer any nitrogen demand from crops?

p13, l259-263: I think it highly unlikely, from a thermodynamic viewpoint, that freezing conditions would boost $NH_3$ emissions from green areas. In cold conditions the Henry coefficient will not favour a shift to the gas phase, but to the condensed phase, and cold temperatures also reduce (micro-)biological activities. I don't actually recall that Flechard et al. (2013) made the argument that higher $NH_3$ emissions from grasslands could be expected under freezing conditions. However, higher $NH_3$ concentrations may occur in the atmosphere in very cold weather for two reasons, i) surface/canopy resistance to dry deposition is higher for a frozen surface, and thus the dry deposition sink strength is reduced and the atmospheric lifetime of $NH_3$ is higher, and ii) if cold weather is associated with a shallow boundary layer and stable conditions (temperature inversion) then $NH_3$ accumulates in the boundary layer at the Earth's surface.

p15, l303: Was there really a significant and steady temperature increase in Downtown Toronto over the 10-year measurement period, that could explain the increasing $NH_3$ trend? How large was the corresponding temperature trend, what was the interannual mean temperature range? From Fig. 4 (right-hand Y axis of upper panel) it looks as though the minimum annual mean temperature was around 8°C, and the maximum value was around 13°C, ie an interannual range of around 5°C, which seems rather large. According to the website weatherstats.ca, the mean annual temperatures in Toronto only ranged from 8 to 10.5 °C during the period 2003-2013 (see below), with no systematic or significant upward trend. I therefore wonder about the accuracy of the temperature data used in producing Fig.4: is it likely that the Toronto mean annual temperature may have been as high as 13°C over that period? In my view this casts some doubt over the argument that an increasing temperature trend was responsible for the increasing $NH_3$ trend. I would encourage the authors to double-check the time series of annual mean temperatures and to show the data in the revised version alongside the mean annual $NH_3$ data.

[Figure]

However, I find the argument of the decreasing SO2 emission more convincing, for two reasons: a decreasing SO2 concentration would lead to less NH3 uptake by acidic (sulphate) aerosols, which is duly mentioned in the paper, but also because less SO2 dry deposition would make the surface less acidic, or more alkaline, which could increase the surface resistance for $NH_3$ (eg Fowler et al., Atmos. Chem. Phys., 15, 13849–13893, 2015, www.atmos-chem-phys.net/15/13849/2015/).

p19, l375-376: "... the long-term change in ambient T possibly dominated the long-term trend in atmospheric NH3 at the site." Again, I don't deny that there is a strong positive correlation between ambient NH3 and temperature on a seasonal or annual basis, or between sites across a continental temperature gradient; this has been shown elsewhere many times, and the reasons for this are mainly thermodynamics and biological. However, what I don't really believe is that there was such a large and systematic increasing temperature trend from year to year over the 10-yr time period considered.

I do however agree that over the long term, climate change and the forecast temperature increases of a few °C will likely result in increased emissions and atmospheric concentrations (See Sutton et al., 2013), but the present dataset is unlikely to show this conclusively, the noise in the signal being likely too high.

Conclusion

p21, l437 onwards: I expect that the "significant decreases in anthropogenic NH3 emissions from main sectors" were calculated on the basis of activity data and related emission factors, which are notoriously highly uncertain and which do not necessarily reflect the true impacts of meteorology and other controls on the NH3 emission processes, if at all. We could therefore argue that the large expected decreases in

agricultural NH3 emissions (as shown in Fig.3) may not necessarily have occurred to the extent they were supposed to. For example, some supposedly "low-emission" slurry spreading techniques that have been introduced over the last 20 years (in Europe, but probably also in N. America?) may not be that efficient after all, and there are also strong methodological issues to be examined regarding the published emission factors (eg Sintermann et al., 2012, http://www.biogeosciences.net/9/1611/2012/bg-9-1611-2012.pdf). The issue of the Dutch ammonia gap, mentioned at the start of the paper, may in part have been a consequence of large uncertainties in NH3 emission inventories and their temporal evolution over the last 20 years.

p21, l440: in addition to changes in biogenic emissions (which may or may not have happened as a consequence of changes in temperatures, as discussed above) and to the changed gas/aerosol NH3/NH4+ partitioning, I would add the possible decrease in NH3 dry deposition rates caused by lower SO2 deposition rates and impact on surface chemistry (eg Fowler et al., Atmos. Chem. Phys., 15, 13849–13893, 2015, www.atmos-chem-phys.net/15/13849/2015/)

**Technical corrections**

In the methods section, there should be two sub-sections, 3.1 NH3 concentration measurements, and 3.2 Statistical methods

p3, l43-44: suggest change to "...the long-term trend in atmospheric NH3 observed in some countries **didn't reflect the** dramatic decrease in NH3 emissions..."

p6, l89: suggest change to "... compiled from three data sources, i.e., the **Canadian** National Air Pollution Surveillance (NAPS,..."

p6, l91: suggest change to "... (CAPMoN), and the **U.S.** Passive Ammonia Monitoring Network..."

p7, l127: change to "...were **referred** to as Site 7-14..."    (not "*refereed*")

p8, l135-137: change to "... Campata et al., 2008). Considering **that** the data flaws aformentioned were indeed **present** in our selected datasets to different extents, the M-K analysis..."

p9, l167: change "*Whitehand*" to "**Whitehead**"

p11, l217: "**Texas**", not "*Taxes*"

p13, l243: "...was much smal**ler**..."

p17, l347: I believe the figure referred to is **Fig S3c**, not *S3b* ?

p20, l417: "...This increase **alone** is not enough to explain..."

p21, l431-432: "respectively, across  North America..."

In Fig S3 (all three panels) in the supplement, change "*Argriculture*" to "**Agriculture**"

**Figures and Tables**

In Figures 2 and 5, and also in Figs S1 and S2, the letters "a,b,c,d, e,f,g,h" in each panel should be changed to Site 1, Site2, .. Site 14

Figure 1: Rather than the split American/Canadian, it would be useful to differentiate and identify sites on the basis of the NAPS, CAPMON and AMON split

For clarity, it would be better to merge Figs 2 and 5,   and also Figs 3 and 6, to show both the measured data and the derived trend lines on the same figures.

Table 1: For each site please provide Lat, Long, Elevation, name of network (NAPS, CAPMON or AMON), land use (urban, agric, remote)

---

## Referee Comment (RC2) · Anonymous Referee #2 · 12 Jul 2016

**Title:** Trends in atmospheric ammonia at urban, rural and remote sites across North America

**Journal:** Atmospheric Chemistry and Physics

**Manuscript Number:** acp-2016-259-manuscript-version2

**Comments:**

This is an interesting paper, in which the authors attempted to compile a large set of measurement data of gaseous ammonia and particulate ammonium at fourteen sites including four urban sites, four remote sites and six rural/agriculture sites distributed at different latitudes in USA and Canada. The authors used two statistical trend analysis techniques to examine trends in atmospheric levels of ammonia over time. The analysis on the role of temperature on ammonia levels is also addressed in this work. Overall, this is a good work that will be beneficial to the global scientific community to get the insights into the trends of atmospheric levels of ammonia. However, the article needs to address some minor issues as per the specific comments given in this report i.e. minor revisions are needed before this paper can be accepted for publication.

- In the manuscript, it was found that strong scientific inferences are stated without support of proper literature citation. E.g., L206-209, L209-211. The authors are requested to fix these kind of issues throughout the manuscript.
- At several places of the manuscript, it is mentioned in a bracket that figure not shown. It would be better to remove such words.
- L303-305: The analysis of data with proper interpretation should be done to support such statement.

---

## Author Comment (AC1) · 26 Aug 2016

Response to Reviewer 1

We greatly appreciate the reviewer for providing the comments which have improved the paper. We have addressed all the comments carefully, as detailed below.

Reviewer's comments on ACPD-2016-259 manuscript "Trends in atmospheric ammonia at urban, rural and remote sites across North America" by X. Yao and L. Zhang

General comments

The paper presents an analysis of temporal trends, and to some extent spatial pat-

terns, of long-term (>10 years) ambient atmospheric ammonia (NH3) concentrations measured across Canadian and US air pollution monitoring networks. Temporal trends are compared between a selection of 14 urban, rural and remote sites by using statistical trend analysis tools (Mann-Kendall, M-K and Ensemble Empirical Mode Decomposition, EEMD). The paper provides a useful and original study of NH3 trends at the N. American continental scale, and fits well within the scope of Atmospheric Chemistry and Physics, even though one may deplore the fact that not all sites and data available in the networks were analyzed.

Long-term trends are interpreted in terms of changing emission patterns and changing pollution climate and temperature, but some of the arguments and hypotheses are less than compelling. In particular the argument that increasing NH3 concentrations at some sites may be explained by a significant upward trend in mean temperatures on such short time scales - from a climatological viewpoint (only 10 years) - does not sound convincing, especially since no long-term temperature data are shown alongside the NH3 concentration time series.

Generally the paper could be improved by a better description of the methods used, both in terms of measurement techniques and statistical methods, and the figures should be re-arranged to combine the actual measured time series with the trends analysis to better illustrate the arguments.

Response: We have added more descriptions of the methods used in this study. We have also incorporated the original data and extracted trends together for better illustrating the points made in this study. According to Walker et al. (Atmos. Environ., 34, 3407–3418, 2000), a seven-year continuous record of data meets the minimum requirements for trend analysis, although longer time periods would be better. Therefore, only the sites met the data requirement were chosen in this study. We agree that with the 7-11 years of data, the climatological trend may not be seen clearly; this is why we used the term "inter-annual variations" in many places. We, however, tried to maximize the knowledge and generated some trends by using the limited data set. We

have adjusted our statements accordingly in the revised paper, keeping in mind such limitations.

Correlations between NH3 concentrations and ambient temperature have been shown in the revised Figures S2 and S3. All extracted trends in NH3 have been shown in Figures 2 and 3. Some processing results for trend analysis have been shown in Figures S5 and 6. As an example analysis, Figure 4 shows the correlation between the extracted trends in NH3 and ambient temperature at Site 1. More correlation results between the two extracted trends have been detailed in the text. To avoid duplication, we decided not to include a figure for the time series of the two variables.

Specific comments

Methods p6, l96-100: please provide more details of the sampling and measurement techniques used in NAPS and CAPMON: which PM2.5 sampler is used (name/manufacturer), are the denuders wet or dry, what is the sample flow rate, are the data hourly or daily integrated values, how is NH4+ measured in the lab after extraction, or is it in-situ online analysis, etc...??

Response: We have added the following description in the revised text: "At each site, a Partisol Model 2300 sequential speciation samplers (Thermo Scientific) equipped with dry denuders is used to measure concentrations of NH3 and acidic gases and particulate chemical components such as pNH4+ and pNO3- in PM2.5. The sampler maintains at a constant flow rate of 10 L min-1 and operates for a 24-hr duration on every third day. The 24-hr integrated denuder and filter samples were carried back to the lab where they were extracted and analyzed by ion chromatography (Dabek-Zlotorzynska et al., 2011)."

p6, l100-107: I agree that missing data are an important problem when dealing with the analysis of long term temporal trends, especially if the downtime periods are not randomly distributed but might tend to coincide with specific weather patterns, eg very cold or very wet, etc. Thus it would be useful to indicate the monthly/annual data

capture rates (eg number of days per month of available data, or rates of missing data, whichever ) alongside the measured concentrations in Figs. 2-3, on a separate axis with a different color or symbol. For example show the missing data rate as a vertical bar for each month, so the figure wouldn't be too cluttered.

Response: Including such information in Figure 2 makes the figure too crowded to be readable. We thereby have added the revised Figure S1 to show the requested information.

p8, l133-141: please provide very briefly the mathematical basis of the statistical method (in which way does it differ from a parametric procedure?)

Response: We have added the following description in the revised paper: "The M-K analysis is a non-parametric statistical procedure which can be used to analyze trends in data sets including irregular sampling intervals, data below the detection limit, and trace or missing data (Kampata et al., 2008). Considering that the data flaws aforementioned were indeed present in our selected datasets to certain extents, the M-K analysis is thereby used to resolve the time series of the annual average of NH3 in this study. According to the analysis, the time series of n data points and Ti and Tj as two subsets of data where i = 1, 2, 3, . . . n-1 and j = i+1, i+2, i+3, . . . n are considered. The series of data is used to calculate S" statistic, confidence factor and coefficient of variation. The calculated values were further used to test the null hypothesis H0 assuming that there is no trend and the alternative hypothesis H1 assuming that there is a trend, yielding qualitative trend results such as "increasing/decreasing", "probable increasing/decreasing", "stable" and "no trend" (Gilbert, 1987)."

Results and discussion p10, l189-193: is there any actual evidence from on-site observations that fertilization takes place in the fall at or around Site 3? Why should there be any fertilizer application after harvest and just before winter, when there is no longer any nitrogen demand from crops?

Response: The record at 74 sites across southern Ontario in 2006 did show fertilization

application in fall (Yao and Zhang, 2103). This is consistent with the report "Canadian Atmospheric Assessment of Agricultural Ammonia, National Agri-Environmental Standards, Environment Canada, Gatineau, Quebec, 2009." Fertilizer application in the late-fall (mid- to late October) is very common in Canada. Fertilizer is applied at the time when the turf has stopped growing but is still green. Such a practice can ensure early growing in the following spring. With the late-fall fertilizer application, spring fertilization can be postponed until late May to early June. Some explanations have been added in the revised paper.

p13, l259-263: I think it highly unlikely, from a thermodynamic viewpoint, that freezing conditions would boost NH3 emissions from green areas. In cold conditions the Henry coefficient will not favour a shift to the gas phase, but to the condensed phase, and cold temperatures also reduce (micro-)biological activities. I don't actually recall that Flechard et al. (2013) made the argument that higher NH3 emissions from grasslands could be expected under freezing conditions. However, higher NH3 concentrations may occur in the atmosphere in very cold weather for two reasons, i) surface/canopy resistance to dry deposition is higher for a frozen surface, and thus the dry deposition sink strength is reduced and the atmospheric lifetime of NH3 is higher, and ii) if cold weather is associated with a shallow boundary layer and stable conditions (temperature inversion) then NH3 accumulates in the boundary layer at the Earth's surface.

Response: The original description is misleading. We have revised the discussion based on this comment. It now reads: "Flechard et al. (2013) also reported higher NH3 mixing ratios over a grassland under freezing condition, which could be due to slower dry deposition process and/or shallow boundary layer."

p15, l303: Was there really a significant and steady temperature increase in Downtown Toronto over the 10-year measurement period, that could explain the increasing NH3 trend? How large was the corresponding temperature trend, what was the interannual mean temperature range? From Fig. 4 (right-hand Y axis of upper panel) it looks as though the minimum annual mean temperature was around 8°C, and the maximum value was around 13°C, ie an interannual range of around 5°C, which seems rather large. According to the website weatherstats.ca, the mean annual temperatures in Toronto only ranged from 8 to 10.5 °C during the period 2003-2013 (see below), with no systematic or significant upward trend. I therefore wonder about the accuracy of the temperature data used in producing Fig.4: is it likely that the Toronto mean annual temperature may have been as high as 13°C over that period? In my view this casts some doubt over the argument that an increasing temperature trend was responsible for the increasing NH3 trend. I would encourage the authors to double-check the time series of annual mean temperatures and to show the data in the revised version alongside the mean annual NH3 data. y = 0.0964x - 184.57 R2 = 0.1543 Annual Mean Temperature Toronto (°C) T_Annual Linear(T_Annual) http://toronto.weatherstats.ca/charts/temperature-25years.html However, I find the argument of the decreasing SO2 emission more convincing, for two reasons: a decreasing SO2 concentration would lead to less NH3 uptake by acidic (sulphate) aerosols, which is duly mentioned in the paper, but also because less SO2 dry deposition would make the surface less acidic, or more alkaline, which could increase the surface resistance for NH3 (eg Fowler et al., Atmos. Chem. Phys., 15, 13849–13893, 2015, www.atmos-chem-phys.net/15/13849/2015/).

Response: The annual average temperature (T) shown in Figure 4 was calculated from the daily averaged values of T on the sampling days when valid NH3 data was available, not every day of the year, and thus might be different from the actual annual average T. Comparing our values shown in Figure 4 with those obtained from the website (http://toronto.weatherstats.ca/), we found general consistence between the two data sets in most of years, with difference less than 0.5 ïČřC in six years and ranged from 1.4 to 2.5ïČřC in the remaining years. The interannual range in annual average T shown in Figure 4 is from 8.0°C to 12.9°C, which is close to the range of 8-10.5°C provided in the website. We agree with the reviewer that the moderately good correlation between the annual average values of NH3 and ambient temperature might be coincident. This is also supported by the decreasing correlation when the EEMD-extracted

results were used for calculation. We agree that the argument of the decreasing SO2 emission appears more convincing. We have revised the discussion accordingly and added related references.

p19, l375-376: "... the long-term change in ambient T possibly dominated the long-term trend in atmospheric NH3 at the site." Again, I don't deny that there is a strong positive correlation between ambient NH3 and temperature on a seasonal or annual basis, or between sites across a continental temperature gradient; this has been shown elsewhere many times, and the reasons for this are mainly thermodynamics and biological. However, what I don't really believe is that there was such a large and systematic increasing temperature trend from year to year over the 10-yr time period considered. I do however agree that over the long term, climate change and the forecast temperature increases of a few °C will likely result in increased emissions and atmospheric concentrations (See Sutton et al., 2013), but the present dataset is unlikely to show this conclusively, the noise in the signal being likely too high.

Response: We agree that long-term trend in T should not increase by more than a few degrees within a decade period; however, interannual variations in T can be higher than a few degrees since there are hot and cool years. Inetrannual variation is a more appropriate term than the long-term trend. We agree that uncertainties exist in correlation analysis and longer measurements are needed for better analysis. We have added this statement in the revised paper "These results suggested that the long-term change in ambient T possibly affected the long-term trend in atmospheric NH3 at the site. However, longer measurements are still needed to confirm the trends in atmospheric NH3 and ambient T as well as their relationship at the site because of the inherent weaknesses in this datasets, such as only seven-years and only one 24-hr sample for every three days available data, and some missing data during this period."

Conclusion p21, l437 onwards: I expect that the "significant decreases in anthropogenic NH3 emissions from main sectors" were calculated on the basis of activity data and related emission factors, which are notoriously highly uncertain and which do not

necessarily reflect the true impacts of meteorology and other controls on the NH3 emission processes, if at all. We could therefore argue that the large expected decreases in agricultural NH3 emissions (as shown in Fig.3) may not necessarily have occurred to the extent they were supposed to. For example, some supposedly "low-emission" slurry spreading techniques that have been introduced over the last 20 years (in Europe, but probably also in N. America?) may not be that efficient after all, and there are also strong methodological issues to be examined regarding the published emission factors (eg Sintermann et al., 2012, http://www.biogeosciences.net/9/1611/2012/bg-9-1611-2012.pdf). The issue of the Dutch ammonia gap, mentioned at the start of the paper, may in part have been a consequence of large uncertainties in NH3 emission inventories and their temporal evolution over the last 20 years.

Response: Agree. The possibility has been added in the revised manuscript to make our analysis more comprehensive. We have also revised our conclusions accordingly to reflect the potential uncertainties in the NH3 emission inventory. Lines 375-380, "However, Sintermann et al., (2012) reported that NH3 emission factors highly depend on meteorology and efficiencies of technical controls on NH3 emission processes. NH3 emission factors adopted in NH3 emission inventories may suffer from uncertainties to some extent, which in turn may transfer to the estimated NH3 emission and affect its relationship with the observed atmospheric NH3." Lines 477-479, it reads "Uncertainties in the NH3 emission inventory are still a concern and may affect the trend analysis results of the relationship between atmospheric NH3 and NH3 emissions."

p21, l440: in addition to changes in biogenic emissions (which may or may not have happened as a consequence of changes in temperatures, as discussed above) and to the changed gas/aerosol NH3/NH4+ partitioning, I would add the possible decrease in NH3 dry deposition rates caused by lower SO2 deposition rates and impact on surface chemistry (eg Fowler et al., Atmos. Chem. Phys., 15, 13849–13893, 2015, www.atmos-chem-phys.net/15/13849/2015/)

Response: We agree with this point and have added a brief discussion and related
references in the revised paper.

Technical corrections In the methods section, there should be two sub-sections, 3.1 NH3 concentration measurements, and 3.2 Statistical methods

Response: Revised as recommended.

p3, l43-44: suggest change to "...the long-term trend in atmospheric NH3 observed in some countries didn't reflect the dramatic decrease in NH3 emissions..." p6, l89: suggest change to "... compiled from three data sources, i.e., the Canadian National Air Pollution Surveillance (NAPS,..." p6, l91: suggest change to "... (CAPMoN), and the U.S. Passive Ammonia Monitoring Network..." p7, l127: change to "...were referred to as Site 7-14..." (not "refereed") p8, l135-137: change to "... Campata et al., 2008). Considering that the data flaws aformentioned were indeed present in our selected datasets to different extents, the M-K analysis..." p9, l167: change "Whitehand" to "Whitehead" p11, l217: "Texas", not "Taxes" p13, l243: "...was much smaller..." p17, l347: I believe the figure referred to is Fig S3c, not S3b ? p20, l417: "...This increase alone is not enough to explain..." p21, l431-432: "respectively, across the North America..."

Response: All corrected.

In Fig S3 (all three panels) in the supplement, change "Argriculture" to "Agriculture" Figures and Tables In Figures 2 and 5, and also in Figs S1 and S2, the letters "a,b,c,d, e,f,g,h" in each panel should be changed to Site 1, Site2, .. Site 14 Figure 1: Rather than the split American/Canadian, it would be useful to differentiate and identify sites on the basis of the NAPS, CAPMON and AMON split For clarity, it would be better to merge Figs 2 and 5, and also Figs 3 and 6, to show both the measured data and the derived trend lines on the same figures. Table 1: For each site please provide Lat, Long, Elevation, name of network (NAPS, CAPMON or AMON), land use (urban, agric, remote)

Response: All corrected and revised accordingly.

---

## Author Comment (AC2) · 26 Aug 2016

Response to Reviewer 2

We greatly appreciate the reviewer for providing the comments which have improved the paper. We have addressed all the comments carefully, as detailed below.

Title: Trends in atmospheric ammonia at urban, rural and remote sites across North America Journal: Atmospheric Chemistry and Physics Manuscript Number: acp-2016-259-manuscript-version2

Comments: This is an interesting paper, in which the authors attempted to compile a large set of measurement data of gaseous ammonia and particulate ammonium at

fourteen sites including four urban sites, four remote sites and six rural/agriculture sites distributed at different latitudes in USA and Canada. The authors used two statistical trend analysis techniques to examine trends in atmospheric levels of ammonia over time. The analysis on the role of temperature on ammonia levels is also addressed in this work. Overall, this is a good work that will be beneficial to the global scientific community to get the insights into the trends of atmospheric levels of ammonia. However, the article needs to address some minor issues as per the specific comments given in this report i.e. minor revisions are needed before this paper can be accepted for publication.

In the manuscript, it was found that strong scientific inferences are stated without support of proper literature citation. E.g., L206-209, L209-211. The authors are requested to fix these kind of issues throughout the manuscript.

Response: This part (L206-211 in original version) has been revised as: "The small seasonal variations in NH3 mixing ratio at this site were caused by two contrasting factors in winter season. On one hand, extremely low temperature limited soil/vegetation NH3 emissions to a level close to negligible (Zhang et al, 2010), as was also seen at Site 1 at ambient temperature <-9°C (Hu et al., 2014). On the other hand, NH3 emissions from industrial and/or non-industrial anthropogenic sources seemed to be enhanced in winter (Lillyman et al., 2009; Behera et al., 2013), as was supported by the two to four times higher mixing ratios of SO2, HONO and HNO3 in winter than in summer."

We have checked through the whole manuscript and revise these accordingly. Please see highlighted parts in the revision.

At several places of the manuscript, it is mentioned in a bracket that figure not shown. It would be better to remove such words.

Response: Deleted.

L303-305: The analysis of data with proper interpretation should be done to support such statement.

Response: The correlations between atmospheric NH3 and ambient temperature and between their long-term trends are presented in Sections 3.1 and 3.3. The relationships between NH3 emissions and ambient temperature and between NH3/NH4+ partitioning and ambient temperature have been well documented in literature. Here, we just summarized those findings and analyzed multiple factors which could affect the observed trends in atmospheric NH3.

To avoid confusion, we have also revised the sentence as " Increasing T not only increases soil/vegetation NH3 emissions but also favors more NH3 partitioning in the gas phase (Pinder et a., 2012; Sutton et al., 2013), both processes would increase NH3 mixing ratios."